# Management of Double-Seropositive Anti-Glomerular Basement Membrane and Anti-Neutrophil Cytoplasmic Antibodies with 100% Crescentic Glomerulonephritis and Nephrotic Range Proteinuria in a Young Female

**DOI:** 10.3390/biomedicines12040906

**Published:** 2024-04-19

**Authors:** Lalida Kunaprayoon, Emily T. C. Scheffel, Emaad M. Abdel-Rahman

**Affiliations:** 1Division of Nephrology, University of Virginia, Charlottesville, VA 22903, USA; vvz3en@uvahealth.org; 2School of Medicine, University of Virginia, Charlottesville, VA 22903, USA; etc3jf@uvahealth.org

**Keywords:** anti-glomerular basement membrane, anti-neutrophil cytoplasmic antibodies, crescentic glomerulonephritis, nephrotic range proteinuria, rituximab

## Abstract

Nephrotic range proteinuria in the setting of dual-positive anti-glomerular basement membrane (AGBM) and anti-neutrophil cytoplasmic antibodies (ANCAs) is rare. Furthermore, using rituximab as a primary immunosuppressant along with steroids and plasmapheresis has not been widely studied. We present a case of dual AGBM and ANCA with nephrotic range proteinuria in a young female, where rituximab was used as a primary immunosuppressant with partial recovery.

## 1. Introduction and Background

Anti-glomerular basement membrane (anti-GBM) disease is a rare rheumatologic glomerulonephritis that is often characterized by progression to rapidly progressive glomerulonephritis (RPGN). Anti-GBM disease occurs in a bimodal age distribution with the first peak between the ages of 15 and 30 years old and a second peak occurring after age 60. Most reported cases are of Caucasian and Asian ancestry [1,2,3].

Anti-GBM is a potentially fatal vasculitis that involves antibodies targeted against the vasculature of alveolar and glomerular basement membranes. As many as 20–40% of patients presenting with anti-GBM are found to have “double-positive” serologies with both anti-GBM antibodies and anti-neutrophil cytoplasmic antibodies (ANCAs), with myeloperoxidase anti-neutrophil cytoplasmic antibodies (MPO-ANCA) specificity being more commonly associated [1,2,3]. Rutgers et al. reports the incidence of double-positive serology with both anti-GBM antibodies and ANCAs to be 0.6 per 1 million population [4] annually, while De Zoysa et al. cited an incidence of 0.47 cases per 1 million population annually [5]. While one antibody may be detected before the other, both ANCAs and anti-GBM antibodies may appear concomitantly [6].

As high as 40–60% of anti-GBM cases will have concurrent lung hemorrhage with similar occurrence in both double- or single-positive anti-GBM disease. Additionally, patients with double-positive antibodies often had extrarenal manifestations, including non-hemorrhagic lower respiratory tract disease, otorhinolaryngological involvement, musculoskeletal symptoms, cutaneous features, neurological, gastrointestinal, and ocular symptoms [1]. Some patients may also present with isolated renal disease [3].

Anti-GBM disease is caused by circulating antibodies produced by B cells, which direct against an antigen intrinsic to the glomerular basement membrane. The principal target for the anti-GBM antibodies is the NC1 domain of the alpha-3 chain of type IV collagen (alpha-3(IV) chain), which is one of six genetically distinct gene products found in basement membrane collagen. The expression of the alpha-3 chain is highest in the glomerular and alveolar basement membranes, lower in the renal tubular basement membranes, and mildly detectable in the choroid plexus, testis, cochlea, and retina, while absent in the small intestine, skin, and placenta [3,7].

In anti-GBM disease, the pathogenic antibodies are usually of the IgG class, predominantly with IgG1 and IgG3 subclasses, although rare cases of IgA- and IgG4-mediated disease have been described. The production of these autoantibodies is thought to be in response to an external inciting factor in genetically susceptible individuals. Anti-GBM disease has a strong HLA-gene association. Patients who inherit HLA-DR2 haplotype with DRB1*1501 and DRB1*0401 alleles appear to have higher risks of developing disease [8]. Several environmental triggers have been reported, including cigarette smoking, hydrocarbon inhalation, and pulmonary infections. The thought was that localized inflammation may disrupt the basement membrane and allow access to pathogenic autoantibodies [3,7]. Alemtuzumab, an anti-CD52 monoclonal antibody used in the treatment of relapsing multiple sclerosis, was identified as a trigger for anti-GBM disease from loss of T cell regulation.

An association between prior kidney injury, such as anti-neutrophil cytoplasmic antibody (ANCA)-associated glomerulonephritis, has been reported, although its mechanism is unclear. Prior study has shown that patients with anti-GBM disease had detectable ANCA prior to the development of anti-GBM antibodies, leading to the clinical manifestations [9]. This suggests that ANCA-associated vasculitis may act as a trigger for anti-GBM disease [7,9].

Studies demonstrated that without treatment for double- or single-positive anti-GBM disease, patient and renal prognosis are both incredibly poor [10]. Established treatment recommendations include plasmapheresis, to rapidly remove pathogenic autoantibody, along with immunosuppression with cyclophosphamide and glucocorticoids, to inhibit further autoantibody production and to mitigate end-organ inflammation [10].

While this treatment regimen has demonstrated vastly improved long-term patient and renal survival, introduction of high-dose gonadotoxic agents like cyclophosphamide presents a challenging dilemma especially in young, previously healthy patients. Such therapeutic decisions are further complicated by the potential use of many other immunosuppressive or immunomodulatory agents that have not yet been proven to be efficacious in double- or single-positive anti-GBM.

One such agent is rituximab, a monoclonal antibody that targets CD20-positive B cells. Its mechanism involves direct induction of apoptosis and antibody- and complement-mediated cytotoxicity. Rituximab has been shown to effectively deplete autoantibodies and induce remission in a variety of glomerular diseases, including ANCA-associated glomerulonephritis [11].

While rituximab has been successful in the management of refractory anti-GBM disease, it has not yet been sufficiently explored as a primary immunosuppressant in double-positive anti-GBM disease [7,12,13].

We present a case of a previously healthy 18-year-old female who presented with RPGN with double-seropositive anti-GBM and p-ANCA with 100% crescentic glomerulonephritis on renal biopsy and nephrotic range proteinuria who demonstrated improved renal function following treatment with plasmapheresis, prednisone, and high-dose rituximab.

## 2. Case Presentation

An 18-year-old female with no significant past medical history presented to an outside hospital with complaints of feeling fatigued for a month followed by generalized abdominal pain and constipation for one week prior to presentation. A month earlier, the patient was treated with amoxicillin for streptococcal pharyngitis. The patient was a high school competitive soccer athlete. Social history revealed no smoking history or illicit substance use and no family history of autoimmune disease or kidney disease.

The patient was found to have non-oliguric acute kidney injury (AKI), hyperkalemia, and metabolic acidosis. Creatinine at presentation was 6.05 mg/dL with an eGFR of 10 mL/min and potassium of 5.0 mmol/L. A urinalysis was significant for red-brown urine with too-numerous-to-count red blood cells and proteinuria. Urine microscopy revealed dysmorphic red blood cells and red blood cell casts. The patient denied any changes to her urine color prior to hospitalization. Extensive laboratory, physical exam, and imaging workup demonstrated no pulmonary, skin, or GI involvement.

The Birmingham vasculitis activity score (BVAS) version 3 was used to evaluate each organ system [14]. The patient’s scores were as follows: General 0, Cutaneous 0, Mucous membranes/eyes 0, ENT 0, Chest 0, Cardiovascular 0, Abdominal 0, Renal 12 (new onset of proteinuria, hematuria, creatinine ≥ 5.65 mg/dL, and 30% rise in serum creatinine), and Nervous system 0. The patient’s total score was 12.

Other laboratory workup was significant for positive anti-GBM and ANCA serologies as well as normal complement (C3 and C4), IgA, creatinine kinase, and liver enzymes (Table 1). Kidney biopsy showed 32 glomeruli. Light microscopy revealed diffuse crescentic glomerulonephritis with 100% mostly cellular with few fibrous crescents and moderate interstitial fibrosis (Figure 1). On immunofluorescence, glomeruli showed linear glomerular basement membrane staining for IgG (+3), kappa (+3), and lambda (+3), with the presence of mesangial and incomplete linear glomerular basement membrane staining for C3 (+2). Electron microscopy revealed severe epithelial foot process effacement with no evidence of an immune complex deposition (Figure 2).

An access for renal replacement therapy (RRT) was placed and the patient was transferred to our institution for initiation of plasmapheresis and hemodialysis. On our initial exam, her vitals were within normal limits and the physical exam demonstrated no abnormalities.

## 3. Management

After consideration of current data and the risks of both rituximab and cyclophosphamide therapies, our team reached a shared decision with the patient and her family to pursue immunosuppressive therapy with only glucocorticoids and rituximab as well as plasmapheresis, foregoing inclusion of the standard cyclophosphamide treatment because of its potential gonad toxicity in a young female 

Glucocorticoids were administered for three days at a dose of 1.0 g methylprednisolone followed by 1 mg/kg/day of oral prednisone through the end of hospitalization. On discharge, the patient was prescribed an 18-week prednisone taper according to the International Society of Nephrology’s (ISN) 2021 Guidelines along with the appropriate antibacterial and proton pump inhibitor prophylaxis treatments.

Daily plasmapheresis was performed for a total of 14 days, starting on hospital day 3, with the goal of continuing until anti-GBM antibodies were no longer detectable in serum. The prescribed treatment was 4 L exchange for 5% human albumin solution. Fresh frozen plasma of 1000 mL was added every 3 days of plasmapheresis to prevent risk of bleeding when fibrinogen level was below 100 mg/dL.

Rituximab was administered twice, each as a 1000 mg dose spaced 14 days apart (on hospital days 5 and 19), with plasmapheresis held for 48 h after each dose of rituximab to decrease washout.

The need for RRT was assessed daily and the patient did require hemodialysis twice, on hospital days 5 and 7.

## 4. Adverse Treatment Effects

The only adverse effects the patient experienced as of this paper’s writing included recurrent allergic reactions during plasmapheresis, leukocytosis, difficulty sleeping, and agitation with increasing steroid doses. The allergic reactions presented as an itchy facial rash soon after initiation of plasmapheresis sessions through an internal jugular line. Administration of IV diphenhydramine was sufficient to control these reactions each time. Additionally, she presented with persistent leukocytosis following initiation of plasmapheresis and these allergic reactions. After a negative infectious workup, it was presumed that this predominantly neutrophilic leukocytosis was likely due to the combinatorial effects of high glucocorticoid doses and reactive leukocytosis from plasmapheresis reactions. This was supported by a steady decline in leukocyte count following cessation of plasmapheresis.

## 5. Discussion and Conclusions

Anti-GBM and double-positive anti-GBM and ANCA glomerulonephritis are rare but potentially fatal autoimmune diseases. Diagnosis of double-positive anti-GBM and ANCA is based on combined analysis of symptoms, serology, and kidney biopsy findings.

Symptomatology is similar in double-positive and single-positive anti-GBM disease, including similar rates of pulmonary involvement and progression to RPGN between the two groups [1,7,15]. While initial renal biopsy often demonstrates higher crescent burden in patients with double-positive disease, overall survival between double- and single-positive anti-GBM cases remains similar [1].

A recent study undertook a systemic review of reported cases with double-seropositive ANCA and anti-GBM. They identified 90 articles in different languages involving patients with double-seropositive ANCA and anti-GBM (*n* = 538). While >90% of the patients had AKI, almost half of them had alveolar hemorrhage. Prognosis was poor in these patients, with overall and renal survival of 64.8% and 38.7%, respectively [6]. Rituximab was used as initial therapy in only 1.6% of their population who were treated with immunosuppressant drugs. Other initial therapies used included cyclophosphamide, which was used in 98.8%, while 4.5% received mycophenolate mofetil, azathioprine, cyclosporin, or other immunosuppressant drugs. Only 61.7% of patients received maintenance therapy. Azathioprine was the drug used the most, with 69% of patients maintained on it, with only 1.7% receiving rituximab for maintenance [6].

A Chinese study analyzed data from patients with double-seropositive patients (*n* = 20) and compared the clinical features and prognosis with patients with MPO-ANCA-associated vasculitis (*n* = 109) and patients with anti-GBM disease (*n* = 23). They demonstrated that double-seropositive patients were older than the other two groups. While serum creatinine was higher in the double-seropositive group than the MPO-ANCA group, it was noted to be lower as compared with the anti-GBM group. No significant difference was noted in the renal and overall survival between the double-seropositive patients and patients with anti-GBM, while patients with MPO-ANCA-associated vasculitis demonstrated a better renal and overall survival compared to the double-seropositive group [16].

Despite such disease severity, there exists little research on treatment for patients in whom the risk profile of cyclophosphamide, the standard immunosuppressive agent, is not an acceptable option. Combination therapy for anti-GBM treatment has been explored with options including cyclophosphamide and mycophenolate mofetil as well as cyclophosphamide and rituximab [7].

It has been accepted that rituximab could be an alternative to cyclophosphamide for ANCA-associated vasculitis. A recent study in patients with single-positive ANCA demonstrated that rituximab treatment is non-inferior to cyclophosphamide-azathioprine treatment in ANCA vasculitis [17]. Upon further review of double-positive ANCA and anti-GBM cases, none have reported the use of rituximab as a primary induction therapy. All cases include glucocorticoids, plasmapheresis, and cyclophosphamide with or without rituximab as an “add-on” therapy [18,19,20,21].

To our knowledge, this is the first case report that shows that patients with double-positive anti-GBM and ANCA may be able to achieve sustainable renal outcome while avoiding the high risk of gonadotoxic side effects from cyclophosphamide.

In their 2021 guidelines [10], the ISN asserted that “A study is needed to compare rituximab to cyclophosphamide, both combined with prednisone plus plasmapheresis for induction of remission.” The guidelines specified scenarios where the use of rituximab is preferred to cyclophosphamide, as in children, adolescents, and pre-menopausal women. They further reported on the success of the use of rituximab in patients with incomplete response to the standard management of anti-GBM glomerular disease. The rarity and severity of double-positive anti-GBM and ANCA disease does not lend itself to the organization of a randomized control trial to study this question. As such, case studies such as the one presented here continue to be imperative drivers in the evolution of anti-GBM treatments.

Another anomalous aspect of this case lies in our patient’s nephrotic range proteinuria. Traditionally, anti-GBM and ANCA vasculitis present with hematuria and sub-nephrotic range proteinuria [22,23]. However, our patient presented with nephrotic range proteinuria, which improved partially following management with steroids and rituximab.

Though rare, a few other case reports have described nephrotic range proteinuria in the setting of anti-GBM disease. Still, with negative testing for membranous nephropathy, focal segmental glomerulonephritis, and no history of NSAID use, the explanation for such a degree of proteinuria remains unclear [22,24,25,26,27].

Nephrotic range proteinuria has been reported in a larger subset of patients with atypical anti-GBM disease, which is defined as renal biopsies consistent with anti-GBM disease without circulating IgG anti-GBM antibodies [26]. Typically, these patients have more indolent disease. The higher incidence of nephrotic range proteinuria in these patients is thought to result from more chronic renal damage and a subsequent higher degree of podocyte injury [26]. A case report has shown an association between double-seropositive ANCA-anti GBM and membranous nephropathy [28], thus accounting for the presence of nephrotic range proteinuria.

Xu et al. reported on nephrotic range proteinuria in patients with ANCA-associated vasculitis. They compared the renal biopsies of the patients with nephrotic range proteinuria (*n* = 20) and compared them with renal biopsies of patients without nephrotic range proteinuria (*n* = 112). They showed that patients with ANCA-associated vasculitis and nephrotic range have more prevalent crescentic glomerulonephritis, higher incidence of AKI, and worse prognosis compared to those without nephrotic range proteinuria [29].

While our patient clearly does not fit into the category of atypical anti-GBM and the renal biopsy did not show any evidence to suggest membranous nephropathy, it could be proposed that, in a similar fashion, her proteinuria resulted from the degree of podocyte injury shown on electron microscopy [22,24,25,26,27]. Her baseline health and high fitness level may have allowed her to maintain daily function for longer after the advent of anti-GBM antibodies, leading to a presentation with higher degree of podocyte and glomerular capillary wall injury and mimicking more chronic or indolent atypical disease courses.

Our case study is certainly not sufficient evidence that rituximab is an equivalent primary immunosuppressive treatment in double-positive anti-GBM and ANCA disease as compared to standard cyclophosphamide treatment. Still, our patient’s unequivocally positive renal response supports our position that rituximab may be a suitable treatment option for patients in whom the risk profile of cyclophosphamide appears unreasonably burdensome. Our treatment elicited a positive response, as the patient was discharged without hemodialysis dependence with a reduction in proteinuria on follow-up. However, her longitudinal prognosis remains uncertain. Long-term follow-up and continued renal monitoring will remain necessary to further understand the efficacy of treatment.

In summary, we present a rare case of a young female with dual-seropositive ANCA and anti-GBM with nephrotic range proteinuria, who had partial response to therapy with steroids, plasmapheresis, and rituximab. Both the presence of nephrotic range proteinuria in this rare entity of ANCA and anti-GBM disease and the management with rituximab with positive renal response, though partial, make this case worth reporting.

## Figures and Tables

**Figure 1 biomedicines-12-00906-f001:**
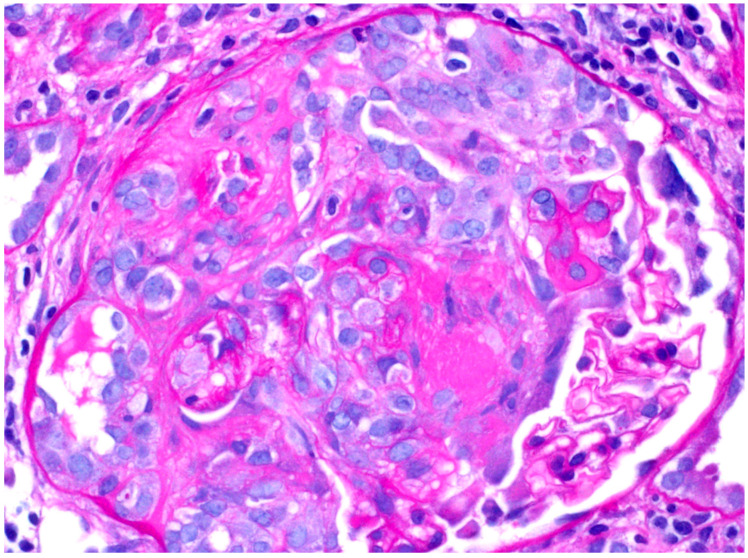
Light microscopy shows cellular crescent formation. Presence of diffuse interstitial edema with mixed interstitial inflammatory infiltrate (Magnification X 400).

**Figure 2 biomedicines-12-00906-f002:**
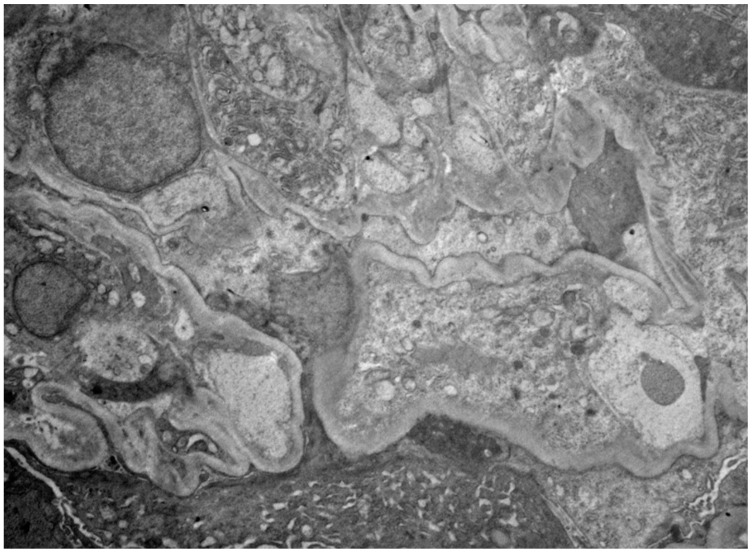
Electron microscopy shows severe epithelial foot process effacement. No evidence of an immune complex deposition observed (x6000).

**Table 1 biomedicines-12-00906-t001:** Laboratory data during and post-admission.

	Day 1	Day 5	Day 9	Day 13	Day 20 (Discharge Day)	2 Weeks Post-Discharge	3 Weeks Post-Discharge
Serum creatinine (mg/dL)	6.05	7.0	4.6	4.0	2.7	3.1	2.4
BUN (mg/dL)	47	59	55	63	56	74	58
K (mmol/L)	5.0	5.6	3.9	4.5	5.1	4.6	3.9
EGFR (mL/min)	10	8	13	16	25	22	29
Anti-GBM (AI) (normal < 1.0)		>8.0		3.2	0.7	1.4	1.3
ANCA Proteinase 3 (AI)		<0.2 (normal)			<0.2 (normal)	<0.2 (normal)	
ANCA, MPO (AI)		6.9			0.3	<0.2 (normal)	
Urine protein/creatinine ratio					12.87	4.25	7.1
C3, C4	Negative						
IgA	WNL						
		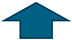		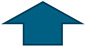			

		Rituximab Days 5 and 19

## Data Availability

Not applicable.

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
