# Peer review of "Management of Double-Seropositive Anti-Glomerular Basement Membrane and Anti-Neutrophil Cytoplasmic Antibodies with 100% Crescentic Glomerulonephritis and Nephrotic Range Proteinuria in a Young Female"

_biomedicines, 2024, doi:10.3390/biomedicines12040906_

Round 1

Reviewer 1 Report

Comments and Suggestions for Authors

This is a case report of crescentic glomerulonephritis with both anti-glomerular basement membrane (GBM) antibody and antineutrophil cytoplasmic antibody (ANCA). In the present case, clinically, the patient's renal function deteriorated to the extent that hemodialysis therapy was required, and histologically, the crescent formation rate was 100%. Despite those, the combination of corticosteroids, rituximab, and plasmapheresis was successful, and this article is considered to be very educational and valuable.

The manuscript was presented in a reasonable manner, but presented clinical features were a little insufficient as described below. Moreover, in my opinion, the authors had better collect previous cases of crescentic glomerulonephritis that were positive for both antibodies and analyze the treatment options.

Below are the points I would like to be added:

1.     Was the involvement of ANCA-associated vasculitis only the renal lesion? Please also provide the Birmingham Vasculitis Activity Score.?

2.     The ANCA antibody titer seems to be relatively low compared to the anti-GBM antibody titer, but is it possible that ANCA plays a major pathological role?

3.     Please provide her serum CRP levels, if examined.

4.     As mentioned above, please summarize the treatment methods for cases with both anti-GBM and ANCA as reported in the literature.

Author Response

Manuscript ID: biomedicines-2954529

Type of manuscript: Case Report

Title: Management of Double Seropositive Anti-Glomerular Basement Membrane and Anti Neutrophil Cytoplasmic Antibodies with 100% Crescentic Glomerulonephritis and Nephrotic-Range Proteinuria in a Young Female.

Authors: Lalida Kunaprayoon, Emily TC Scheffel, Emaad Abdel-rahman MD *

Response to Reviewer 1

Thank you so much for your thorough and thoughtful review. Kindly see our responses below

  1. Was the involvement of ANCA-associated vasculitis only the renal lesion? Please also provide the Birmingham Vasculitis Activity Score.?

** Only the renal lesion was involved in ANCA-associated vasculitis and the Birmingham Vasculitis Activity Score (BVAS) is provided in case presentation.

  1. The ANCA antibody titer seems to be relatively low compared to the anti-GBM antibody titer, but is it possible that ANCA plays a major pathological role?

**Excellent point. There is a possibility as the treatment for double-positive anti-GBM targets both Anti-GBM antibodies and circulating ANCA. As mentioned, undetected ANCA can also be a trigger for Anti-GBM clinical disease presentation.

3.     Please provide her serum CRP levels, if examined.

** CRP was not obtained

4.     As mentioned above, please summarize the treatment methods for cases with both anti-GBM and ANCA as reported in the literature.

 **Added under discussion

Reviewer 2 Report

Comments and Suggestions for Authors

The article presents an interesting case report, where you can see a clinical case of an 18-year-old patient. However, there are different details that should be looked at:

1 - the characterization of the patient, sociological, demographic, family, environmental, ect. In order to infer a link between the patient and her pathology.

2 - we don't understand the follow-up time, because a clinical situation of this nature should be followed up for at least 3 weeks, I'd say monthly, and only end in a year....

3 - Figures 2 and 3 need a scale on the image.

4 - the discussion is very confusing; it needs to be simplified and improved.

Author Response

Manuscript ID: biomedicines-2954529

Type of manuscript: Case Report

Title: Management of Double Seropositive Anti-Glomerular Basement Membrane and Anti Neutrophil Cytoplasmic Antibodies with 100% Crescentic Glomerulonephritis and Nephrotic-Range Proteinuria in a Young Female.

Authors: Lalida Kunaprayoon, Emily TC Scheffel, Emaad Abdel-rahman MD *

Response to reviewer 2

Thank you so much for your thoughtful and thorough review. Below are our responses

1 - the characterization of the patient, sociological, demographic, family, environmental, ect. In order to infer a link between the patient and her pathology.

** There was no significant family history. It has been mentioned that the patient did not smoke, use illicit drugs, or encounter any unusual environmental factors. She is a high school competitive soccer athlete.

2 - we don't understand the follow-up time, because a clinical situation of this nature should be followed up for at least 3 weeks, I'd say monthly, and only end in a year.

**Agree with reviewer. We plan to continue to follow the patient longitudinally. Her disease process started few weeks ago and we included our follow up till the time the manuscript was written.

3 - Figures 2 and 3 need a scale on the image.

**Unfortunately, the biopsy images provided did not include the scale on the image

4 - the discussion is very confusing; it needs to be simplified and improved.

**Done

Reviewer 3 Report

Comments and Suggestions for Authors

Reviewing the manuscript entitled, “Management of Double Seropositive Anti-Glomerular Basement Membrane and Anti Neutrophil Cytoplasmic Antibodies with 100% Crescentic Glomerulonephritis and Nephrotic-Range Proteinuria in a Young Female” by Kunaprayoon L et al., this is a case report regarding the treatment of steroids and rituximab in an 18-year-old female patient with advanced AKD who was double positive for anti-GBM and anti-ANCA antibodies. This is a case in which steroids and rituximab, which are not recommended by guidelines, were relatively successful. This kind of case report may be important because the target is a rare disease and it is extremely difficult to evaluate drug efficacy through randomized clinical trials.

 The patient was an 18-year-old female. Considering gonadal toxicity, plasma exchange, steroids, and rituximab were selected, and the treatment showed some efficacy. However, these are not standard treatments. If so, regarding the possible mechanisms of treatment with these drugs, the authors should cite the manuscript and explain it a little more. An explanation is also required regarding the anti-CD20 monochrome antibody. It would be easier to understand if there was a flowchart for determining therapeutic drug strategy based on personalized medicine.

 Although it is a case report, the authors should show approval from the ethics committee.

Author Response

Manuscript ID: biomedicines-2954529

Type of manuscript: Case Report

Title: Management of Double Seropositive Anti-Glomerular Basement Membrane and Anti Neutrophil Cytoplasmic Antibodies with 100% Crescentic Glomerulonephritis and Nephrotic-Range Proteinuria in a Young Female.

Authors: Lalida Kunaprayoon, Emily TC Scheffel, Emaad Abdel-rahman MD *

Response to reviewer 3

Thank you so much for your thoughtful and thorough review. Below are our responses

Reviewing the manuscript entitled, “Management of Double Seropositive Anti-Glomerular Basement Membrane and Anti Neutrophil Cytoplasmic Antibodies with 100% Crescentic Glomerulonephritis and Nephrotic-Range Proteinuria in a Young Female” by Kunaprayoon L et al., this is a case report regarding the treatment of steroids and rituximab in an 18-year-old female patient with advanced AKD who was double positive for anti-GBM and anti-ANCA antibodies. This is a case in which steroids and rituximab, which are not recommended by guidelines, were relatively successful. This kind of case report may be important because the target is a rare disease and it is extremely difficult to evaluate drug efficacy through randomized clinical trials.

The patient was an 18-year-old female. Considering gonadal toxicity, plasma exchange, steroids, and rituximab were selected, and the treatment showed some efficacy. However, these are not standard treatments. If so, regarding the possible mechanisms of treatment with these drugs, the authors should cite the manuscript and explain it a little more. An explanation is also required regarding the anti-CD20 monochrome antibody. It would be easier to understand if there was a flowchart for determining therapeutic drug strategy based on personalized medicine.

 Although it is a case report, the authors should show approval from the ethics committee.

**Further explanation regarding the purpose of standard treatments and more mechanisms on the use of rituximab was added in the introduction/background.

**The management plan and drug timeline have been extensively shown in detail on Figure 3 and 4.

**Informed Consents have been signed. The Ethics committee was waived as this is a case report.

Round 2

Reviewer 1 Report

Comments and Suggestions for Authors

  This is a case report of crescentic glomerulonephritis with both anti-glomerular basement membrane (GBM) antibody and antineutrophil cytoplasmic antibody (ANCA).

  The revised manuscript was presented in a reasonable manner, and presented clinical features were well described. Authors had also responded to my all comments.

Reviewer 2 Report

Comments and Suggestions for Authors

the authors didn't make my suggestions, so in my opinion it shouldn't be accepted. Although I understand that some of them were complex, other suggestions were very simple, such as the first or third.